# Age and sex-related variations in murine laryngeal microbiota

Ran An[1]☉, Anumitha Venkatraman[1]☉, John Binns[1], Callie Saric[2], Federico E. Rey[3], Susan L. Thibeault[1]*

1 Department of Surgery, School of Medicine and Public Health, University of Wisconsin-Madison, Madison, WI, United States of America, 2 Department of Medicine, School of Medicine and Public Health, University of Wisconsin-Madison, Madison, WI, United States of America, 3 Department of Bacteriology, College of Agriculture and Life Sciences, University of Wisconsin-Madison, Madison, WI, United States of America

☉ These authors contributed equally to this work.
* thibeault@surgery.wisc.edu

**Data Availability Statement:** All of the data is available at GenBank BioProject ID: PRJNA1041457.

**Funding:** The work has been funded by the National Institutes of Health – National Institute on

## Abstract

The larynx undergoes significant age and sex-related changes in structure and function across the lifespan. Emerging evidence suggests that laryngeal microbiota influences immunological processes. Thus, there is a critical need to delineate microbial mechanisms that may underlie laryngeal physiological and immunological changes. As a first step, the present study explored potential age and sex-related changes in the laryngeal microbiota across the lifespan in a murine model. We compared laryngeal microbial profiles of mice across the lifespan (adolescents, young adults, older adults and elderly) to determine age and sex-related microbial variation on 16s rRNA gene sequencing. Measures of alpha diversity and beta diversity were obtained, along with differentially abundant taxa across age groups and biological sexes. There was relative stability of the laryngeal microbiota within each age group and no significant bacterial compositional shift in the laryngeal microbiome across the lifespan. There was an abundance of short-chain fatty acid producing bacteria in the adolescent group, unique to the laryngeal microbiota; taxonomic changes in the elderly resembled that of the aged gut microbiome. There were no significant changes in the laryngeal microbiota relating to biological sex. This is the first study to report age and sex-related variation in laryngeal microbiota. This data lays the groundwork for defining how age-related microbial mechanisms may govern laryngeal health and disease. Bacterial compositional changes, as a result of environmental or systemic stimuli, may not only be indicative of laryngeal-specific metabolic and immunoregulatory processes, but may precede structural and functional age-related changes in laryngeal physiology.

## Introduction

The larynx is essential for voice production. As we age, the larynx undergoes structural and biological changes in the vocal fold mucosa, laryngeal cartilages, and muscle fibers [1–4], which impact voice quality [5–8]. Emerging evidence suggests that laryngeal microbiology can

Deafness and other Communication Disorders
R01DC012773. The funders had no role in study
design, data collection and analysis, decision to
publish, or preparation of the manuscript.

**Competing interests:** The authors have declared
that no competing interests exist.

influence immunological processes [9]. Age-related microbial variation may potentially under-
lie biological mechanisms and leave the larynx more susceptible to disease pathogenesis, as
laryngeal pathology is more prevalent with increasing age [10]. Given the fact that age-related
laryngeal pathology can affect 10–20% of the elderly population [5–8], there is a need to delin-
eate laryngeal microbial changes as a function of age. This could lay the groundwork for
informing age-related host-microbial interactions in laryngeal pathology.

Although studies in the laryngeal microbiota are relatively sparse, age-related microbial var-
iation has been delineated in other organs, including the gut and oral cavity [11, 12]. As we
age, the gut replaces certain short-chain fatty acid (SCFA)-producing bacteria (*Prevotolla*, *Bifi-
dobacterium*) with other phyla that perform the same function, such as *Oscillospora* [11, 12].
This process prevents the host from physiological age-related decline and improves gut barrier
function [11]. Certain bacterial taxa, such as *Veillonella*, *Ruminococcus*, *Akkermansia*, in the
gut induce "inflammaging", the processes wherein there are low levels of systemic inflamma-
tion in older adults [13], which can leave them susceptible to further infection [14]. A decrease
in abundance of *Lactobacillus* and *Bifidobacterium* species can be associated with aging [15,
16]. Age-related microbial dysbiosis may be site specific, as decreased abundance of *Fusobac-
terium* genera has been measured in the gut with aging, while decreased abundance of *Propio-
nibacterium* is seen in the skin and increased *Actinomycoses* in the oral cavity [14]. Oral
microbiota varies highly across the lifespan, with middle ages showing a more homogenous
composition and older ages showing more diverse microbiomes with increased representation
of typically low abundance taxa [17]. The abundance of oral commensal *Neisseria* decreases
after the age of 40; prevalence of opportunistic pathogens, *Streptococcus anginosus* and *Gemella
sanguinis*, exhibit increased abundance with age [18]. In contrast, a longitudinal study of the
human oropharynx microbiota reveals a common core microbiota with high temporal stability
over 40 weeks' observation [19]. While situated proximal to the oral cavity, the larynx, sitting
at the crossroads between the respiratory and digestive tracts, imposes unique selective pres-
sures on resident bacteria. Thus, age-related microbial variation in the larynx should be inves-
tigated independently.

Age-related variations in microbiota can interact with a number of factors, including sex,
diet, and environment. Evidence suggests that sex hormones can potentially alter the gut
microbiota throughout the lifespan [20, 21]. Sexual dimorphism exists in laryngeal muscle
fibers, laryngeal size and manifests as differences in vocal quality [22]. Not only are females
more prone to voice disorders when compared to males [23, 24], but certain pathologies can
affect the sexes disproportionately. Understanding sex-related microbial mechanisms in the
larynx can potentially improve knowledge on certain disease pathogenesis. As a secondary aim
to our study, we explored the effects of biological sex on laryngeal microbiota in the murine
model.

The murine model has been frequently used in the study of laryngeal microbiology and
other biological processes [25, 26]. Thus, we compared age and sex-related variation in laryn-
geal microbial profiles of adolescent, adult, older adults and elderly mice. We found no age or
sex-related differences, indicating the laryngeal microbiota remains relatively stable across the
lifespan. Subtle differences in microbial composition, specifically for the following phyla; *Fir-
micutes*, *Bacteroides*, and *Corynebacterium*, are observed among different age groups.

## Methods

All procedures and protocols were approved by the Institutional Animal Care and Use Com-
mittee (IACUC) at University of Wisconsin-Madison (Protocol: M005669).

## Description of samples and group allocation

Conventionally-raised C57BL/6J mice were divided into the following groups based on their age: adolescent/6Wk (n = 6; age range: 6 weeks +/- 1 day; sex: 3 females (F), 3 males (M)), young adult/6Mo (n = 7; age range: 6 months +/- 19 days; sex: 4F, 3M), older adult/12Mo (n = 6; age range: 12 months +/- 21 days; sex: 3F, 3M), and elderly/18Mo (n = 5; age range: 18 months +/- 7 days; sex: 3F, 2M) [27]. All mice had the same lineage and housing conditions with ad libitum access to regular chow diet and acidified tap water in a 12-hour light/dark cycle throughout their respective lifespans. For the experiment, mice were sacrificed via $CO_2$ inhalation, and larynges were excised with sterile surgical tools for downstream processing.

## Sample processing

To collect bacteria from the laryngeal mucosa, laryngeal tissue was hemisected and minced in a sterile petri dish. Minced tissue was washed 2–3 times in sterile Dulbecco's phosphate-buffered saline (DPBS, Ca/Mg-free) in a 2 mL microcentrifuge tube vortexed on a Genie 2 Vortex Mixer (Scientific Industries Inc., Bohemia, New York, USA) with horizontal microtube holder (LABRepCo, Horsham, Philadelphia, USA), at maximum speed for 5 min. Solid tissues were removed from the cell suspension, and bacterial cells were harvested by centrifuging cell suspension (lavage) at 15,000 rpm for 10 min. Bacterial cell pellets were stored in -80˚C until further processing. Within a week, bacterial cell pellets were thawed, and bacterial DNA from each sample was extracted using DNeasy Blood and Tissue Kit (Qiagen, Hilden, Germany) as per manufacturer's instructions. Extracted DNA was eluted in 35 uL of low TE buffer (10 mM Tris/0.1mM EDTA), and quantified using Qubit® Fluorometer (Invitrogen, San Diego, CA, United States). On average 40 ng/uL of bacterial DNA was obtained from each mouse larynx and stored at 4˚C until 16S rRNA gene amplification.

V4 region of the 16S rRNA gene was amplified for each bacterial DNA sample using DNA-free Platinum Taq Polymerase (Invitrogen, Waltham, Massachusetts). Amplification was completed in a 25 uL PCR reaction that contained 20 ng of DNA template and 400 uM of 515F/806R primers [25]. An extraction negative control, no template control, and positive control (mouse fecal DNA) was included in each test PCR run. Thermocycling parameters of PCR reactions were as follows: initial denaturation (95˚C for 3 min), 35 cycles of denaturation (95˚C for 30 s), annealing (55˚C for 30 s), extension (72˚C for 1 min), followed by final extension (72˚C for 5 min). PCR Amplicons were confirmed on 1.5% agarose gel, and amplicon concentration was subsequently quantified with a Qubit® Fluorometer (Invitrogen, San Diego, CA, United States). Oral Microbiome Whole Cell Mix (ATCC, Manassas, VA) (n = 4) was included as positive control, processed, and sequenced in parallel with tissue samples, to examine the effectiveness of sample/data processing methods. The mock community was a defined synthetic community comprised of equal portions of *Schaalia odontolytica* (ATCC 17982), *Prevotella melaninogenica* (ATCC 25845), *Fusobacterium nucleatum* subsp. *nucleatum* (ATCC 25586), *Streptococcus mitis* (ATCC 49456), *Veillonella parvula* (ATCC 17745), and *Haemophilus parainfluenzae* (ATCC 33392), with each representing 16.7% of total cells. PCR products from the biological replicates and positive control were pooled for each age group into equimolar libraries and identified on a 1.5% agarose gel. A Zymoclean Gel DNA Recovery Kit (Zymo research, Irvine, CA) was used to obtain a purified DNA library (containing 8 ng of DNA/sample). This library was sequenced on Illumina MiSeq platform (Illumina, San Diego, CA) with 250-bp paired-end sequencing chemistry by UW-Madison Biotechnology Center.

## Sequencing and statistical analysis

Demultiplexed sequences were processed using QIIME2 (v2022.11). QIIME2 is a widely used plug-in based platform for microbiome analysis [28]. A DADA-2 pipeline, using the open-

sourced q2-dada2 plugin (v2017.2.0), identified *de novo* amplicon sequence variants (ASVs) following quality-filtering and denoising [29]. ASVs were then aligned with Multiple Alignment using Fast Fourier Transform (MAFFT) using the q2-alignment plug-in (v3.6) [30]. Shared ASVs across groups were subsequently visualized. Unique taxa were identified through *classify-sklearn* against the Greengenes 13_8 99% references sequences (open-source) [31, 32]. Microbial composition at each taxonomic level was obtained with the QIIME2 *taxa-collapse* function.

Measures of alpha-diversity, beta-diversity, and subsequent analyses were obtained using QIIME2 and Rstudio (v3.30), at a rarefaction depth of 531 sequences per sample. Two samples (1 adolescent and 1 young adult) were removed due to the cutoff set by rarefaction depth. For beta-diversity analysis, microbial community differences between different ages and sexes were evaluated with permutational analysis of variance (PERMANOVA) and visualized using principal coordinate analysis (PCoA). Dominant taxa and indicator species [33] for each age group were subsequently graphed using *indicspecies* (v1.7.14) package in R. One-way analysis of variance (ANOVA) and Tukey HSD post-hoc comparisons were incorporated to delineate differential abundances of top 4 genera across age and sex groups. Differential abundances across age groups and sexes were analyzed using linear discriminant analysis (LDA) effect size (LEfSe) using Galaxy/Hutlab open-source software [34]. *limma* (v3.18) package was used to verify the results of differential abundance analysis across sexes [35]. Bacterial taxa with an LDA score $> 4$ and a p value $< 0.05$ were considered enriched for that group. PICRUSt2 (v2.5.2) analysis [36] was completed to determine functional consequences of age and sex on the laryngeal microbiome. Functional predictions were obtained from comparisons with KEGG pathway assignments. Welch's t test comparisons were completed to compare gene and pathway abundances in the laryngeal microbiome of different age and sex groups, and the results were visualized in STAMP (v2.1.0) tool [37].

## Results

### Laryngeal microbial profiles as a function of age

We compared murine laryngeal microbiota profiles using 16 s rRNA sequencing across different age groups [adolescent, mature adult, older adult, and elderly] [38]. After removing ASVs unique to a single sample or present in less than 10% of the samples, we found 85 unique ASVs in laryngeal microbiota. The number of shared ASVs between adjacent age groups varied with age [adolescent and young adult mice share 6 ASVs, young and older adults share 12 ASVs, and older adults and the elderly share no unique ASVs [that are not present in the other groups, Fig 1]. Although all age groups share 14 ASVs, there were no ASVs unique to only the oldest age groups (older adult and elderly, Fig 1). This may be indicative of gradual bacterial compositional changes that may occur as a function of age.

Fig 2 represents taxonomic profiles of the four age groups at phylum (top) and genus (bottom) level. *Firmicutes* and *Actinobacteria* were the two dominant phyla across the age groups, while *Corynebacterium*, *Streptococcus*, and *Lactobacillus* were the top three genera across the age groups. *Lactobacillus* dominated in adolescents, *Corynebacterium* in young adults, and *Streptococcus* in older adults (Fig 2). At the phyla level, mean relative abundance of *Firmicutes* was dominant in adolescents, while *Actinobacteria* and *Proteobacteria* in young adults and elderly (Fig 3). Taxonomic profiles of the top 4 genera across age groups (Fig 4) demonstrated a significant decrease of *Corynebacterium* in the adolescent group, compared to other groups ($p < 0.05$, Fig 4). Adolescents had the largest relative abundance in *Lactobacillus*, with large variation across groups. There was a significant decrease in the relative abundance of *Streptococcus* in the young adult mice when compared to the older adult and elderly ($p < 0.05$, Fig 4).

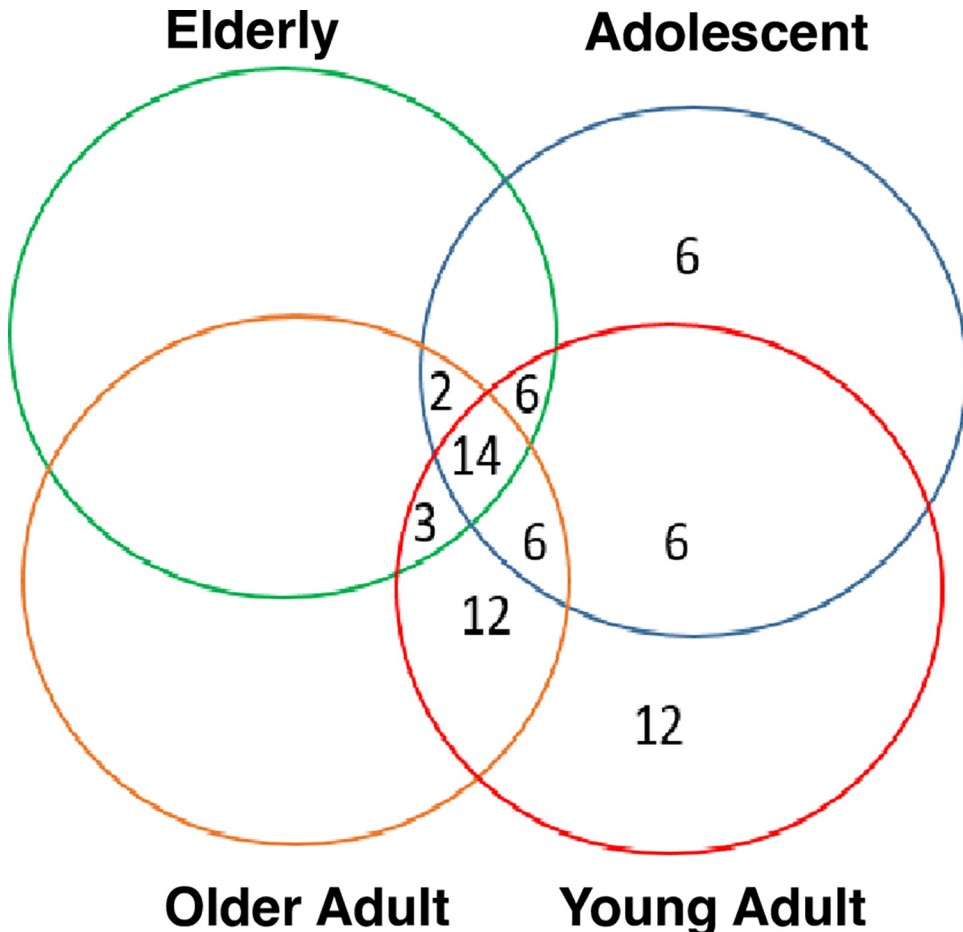

**Fig 1. Amplicon sequence variants in the laryngeal microbiota shared between murine age groups across the lifespan.** After removing ASVs unique to a single sample or present in less than 10% of samples, we found 67 unique ASVs in laryngeal microbiota. The number of shared ASVs between adjacent age groups varied with age [adolescent and young adult mice share 6 ASVs, young and older adults share 12 ASV, and older adults and the elderly share no unique ASVs (that are not present in the other groups), Although all age groups share 14 ASVs, there were no ASVs unique to only the oldest age groups (older adult and elderly).

Lastly, *Bacteroides (S24-7)* was significantly reduced in the young adult group, compared to the adolescent group ($p <$ 0.05, Fig 4). We conducted indicator species analysis to identify genera indicative of age-related variation in the laryngeal microbiota, detailed in Table 1. We further identified differentially abundant taxa across age groups through LEfSe analysis and found *Corynebacterium* was significantly enriched in the young adult group (LDA score > 4, Fig 5). Metagenomic functional profiles predicted by PICRUSt2 based on 16S rRNA gene data revealed 32 KEGG pathways differentially represented across age groups ($p$ = 0.05, Effect size >2, Table 2). These functions were primarily associated with metabolism, cellular processes, and genetic information processing, etc. Laryngeal microbiota of adolescents (6Wk) collectively encode for significantly distinct functions from that of older mice, especially young adults (6Mo) (Fig 6A1). Butyrate and the overall fatty acid metabolism exhibited the highest proportion of sequences in young adult mice (6Mo) (Fig 6B1); propionate metabolism was high in young adult and elderly mice (Fig 6B1). Interestingly, the three metabolisms were lowest in adolescent mice (6Wk) (Fig 6B3).

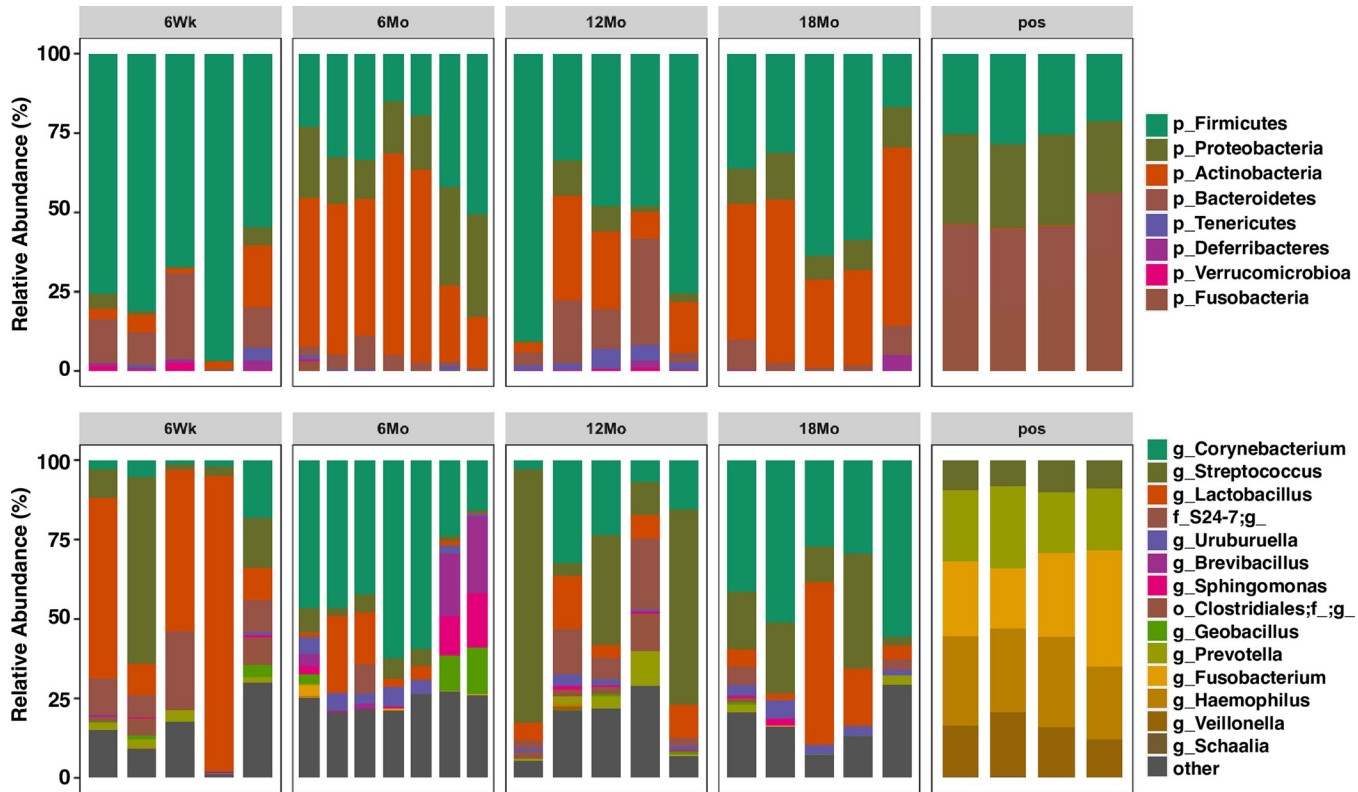

**Fig 2.** Taxonomic compositions of microbiota at phylum (top) and genus (bottom) levels in conventionally raised C57BL/6J mice across age groups. p, g, f, and o represent phylum, genera, family, and order level; other represents the relative abundance of all other phyla or genera combined; 6Wk = adolescent, 6Mo = young adult, 12Mo = older adult, 18Mo = elderly.

There were no significant differences in the number of ASVs (Observed ASV Richness), biodiversity (Shannon's diversity), or evenness (Pielou's evenness) between age groups, indicating minimal variation in bacterial diversity across age groups (Fig 7). Principal coordinates analysis of unweighted UniFrac distances (Fig 8A) did not show significant clustering between age groups ($p > 0.05$); weighted UniFrac and unweighted UniFrac distances (Fig 8B) did not demonstrate significant clustering between age groups with PERMANOVA ($p > 0.05$).

## Laryngeal microbial profiles as a function of biological sex

There were no significant sex differences in relative abundance of the top 4 genera ($p>0.05$, Fig 9). There were no significant differences in the number of ASVs (Observed ASV Richness), biodiversity (Shannon's diversity), or evenness (Pielou's eveness) between sexes, indicating biological sex has little impact on commensal bacterial diversity in the larynx (Fig 10) PCoA of unweighted Unifrac distances (Fig 11A) and weighted Unifrac distances (Fig 11B) did not show significant clustering between sexes. LEfSe analysis exhibited no differentially abundant taxa across biological sex ($p>0.05$, Fig 6A2), which was consistent with the differential abundance analysis using *limma* by fitting the data to a linear model with age as a confounding variable ($p = 0.05$). Thus, no indicator species analysis was completed for the biological sexes. Taken together, this is indicative of minimal taxonomic variation between sexes. Metagenomic functional profiles predicted by PICRUSt2 revealed no KEGG pathways differentially represented between sexes ($p = 0.05$, Effect size = 2).

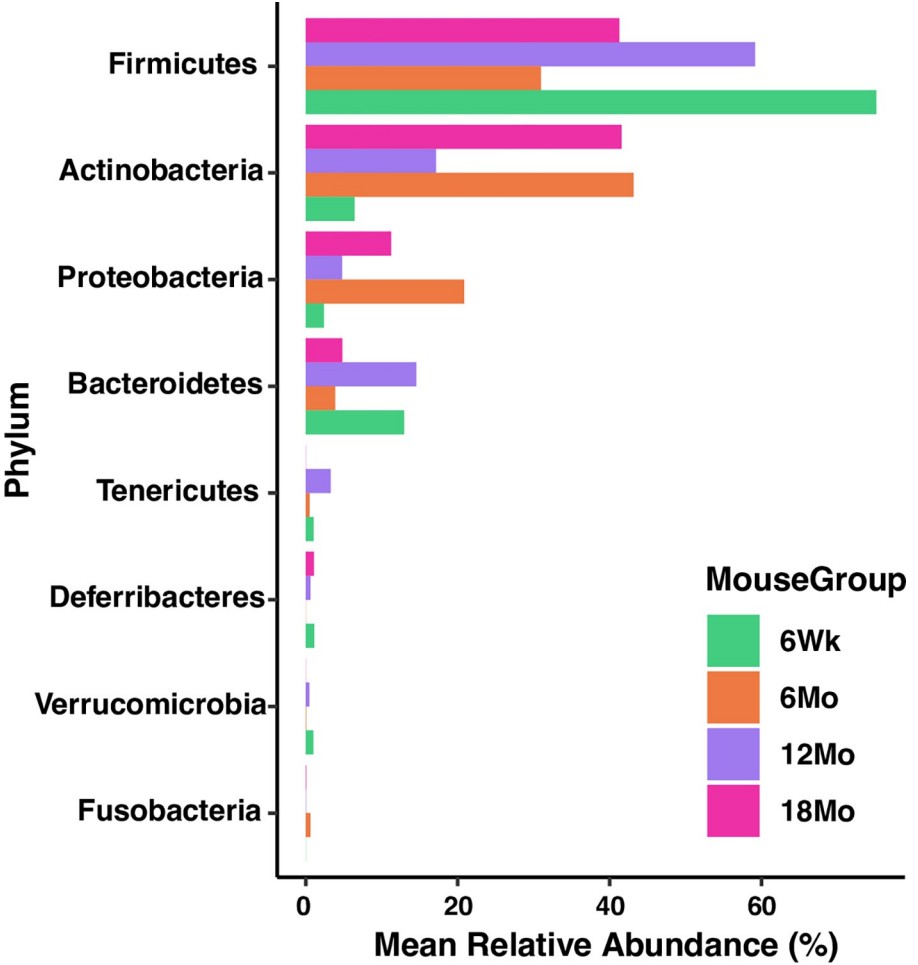

**Fig 3. Dominant phyla in the laryngeal microbiota across various age groups.** 6Wk = adolescent, 6Mo = young adult, 12Mo = older adult, 18Mo = elderly.

## Discussion

Age is an important factor influencing laryngeal structure and function, with a greater number of voice disorders reported in older adults [10]. Age-related microbial variation is complex in other organs (gut, nose and mouth) and can be affected by a number of factors including diet, environment, and systemic conditions [11, 14, 39]. Constant regeneration and repair in the larynx occurs as a result of mechanical, systemic, and environmental influences [40]. Thus, there is a critical need to parse out independent microbial mechanisms that may be mediated by age-related changes within the larynx. The primary aim of our study was to delineate age-related variation in laryngeal microbiota by comparing murine samples throughout the lifespan (adolescent, young adult, older adult, and elderly groups).

At the phyla level, the adolescent murine group was dominated by *Firmicutes*. These bacteria are capable of producing SCFA, such as butyrate. Butyrate is important for breaking down dietary fiber in the gut and acts as an energy source for the body. In the respiratory system, SFCA-producing bacteria are responsible for regulating immunological homeostasis and susceptibility to diseases [41]. Bacteria belonging to *Firmicutes* produce 10% of the body's energy supply, while regulating gut homeostasis [42–45]. In healthy gut aging, SCFA-producing

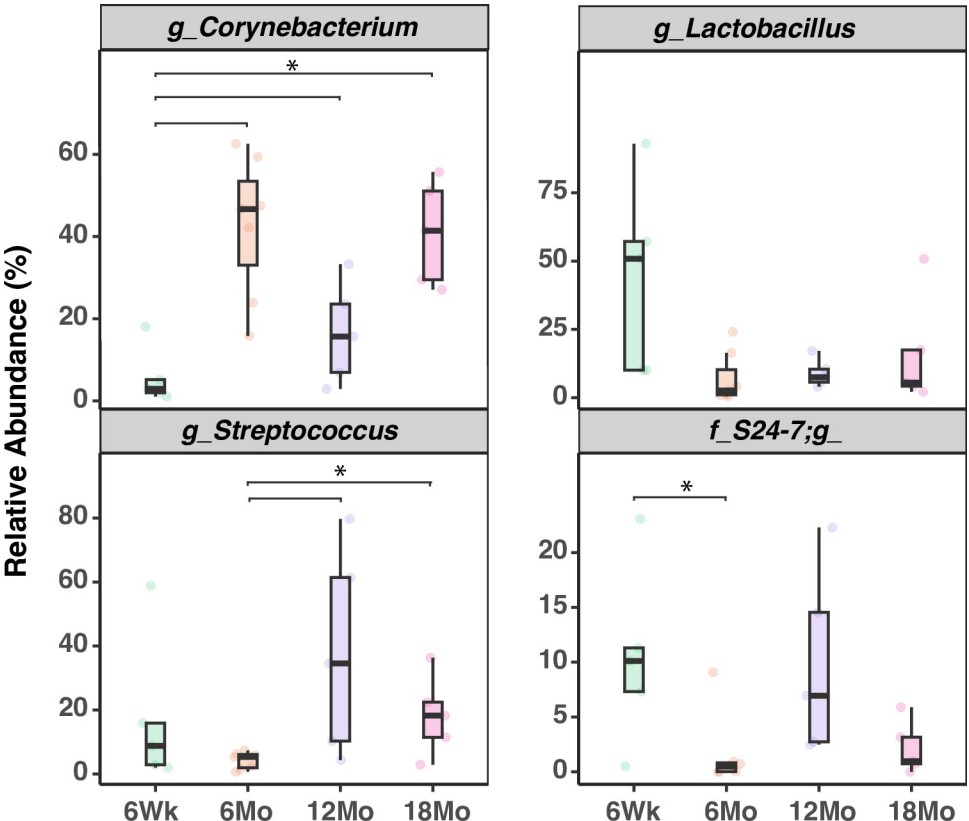

**Fig 4. Comparison of the relative abundance of top 4 genera across age groups.** g, f represent genera and family level. 6Wk = adolescent, 6Mo = young adult, 12Mo = older adult, 18Mo = elderly. Statistical analyses were done by one-way ANOVA and Tukey HSD post-hoc comparisons across age groups; * represents the differentially abundant taxa were statistically significant ($p < 0.05$).

*Firmicutes* is replaced with other SCFA-producing bacteria, such as *Oscillospora* (which can decrease certain inflammatory diseases in the gut) [11, 12]. However, we found that *Oscillospora* were identified as an indicator species in the larynges of adolescents, in addition to the

**Table 1. Indicator species of each age group.**

| Taxonomy (genus) | Group | Stat | p-value |
|---|---|---|---|
| *Ruminococcus* | Adolescent | 0.560 | 0.0425* |
| *Bifidobacterium* | Adolescent | 0.544 | 0.0283* |
| *Lactobaccilus* | Adolescent | 0.537 | 0.04* |
| *Oscillospora* | Adolescent | 0.649 | 0.002** |
| *Akkermansia* | Adolescent | 0.648 | 0.02* |
| *Prevotella* | Adolescent | 0.6348 | 0.02* |
| *Actinomyces* | Young Adult | 0.568 | 0.0357* |
| *Veillonella* | Young Adult | 0.610 | 0.03* |
| *Fusobacterium* | Young Adult | 0.98 | .001** |

Significance code

** 0.01

* 0.05

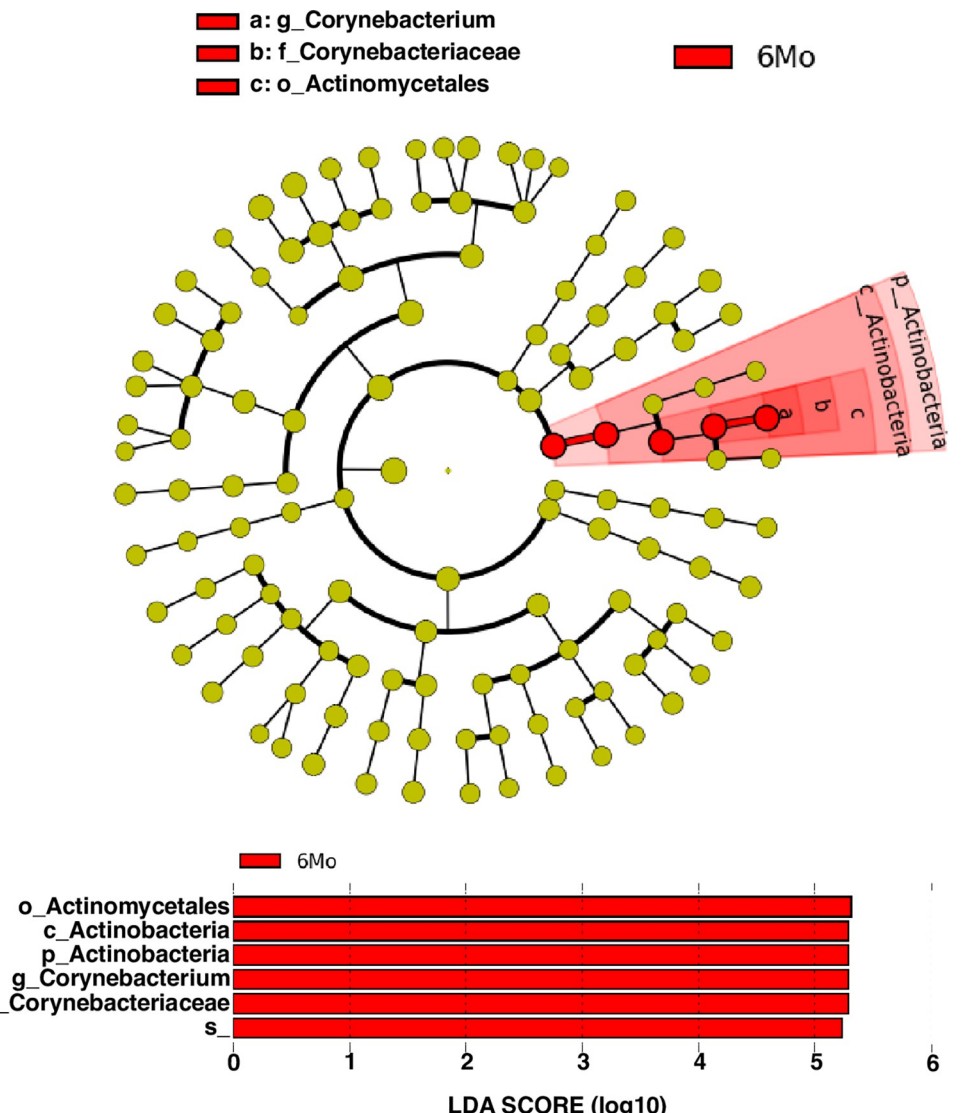

**Fig 5. Differentially abundant taxa across age groups based on LEfSe analysis.** Linear discriminate analysis (LDA) effect size (LEfSe) was performed to determine the differentially abundant taxa across age groups. Cladogram showing the significantly different taxa at different levels across age groups; histogram of LDA scores computed (LDA score threshold = 4) for taxa differentially abundant across age groups. LEfSe score shows the degree of consistent difference in relative abundance for taxa across age groups. g, f, o, c, and p represent genera, family, order, class, and phylum level; 6Mo = young adult.

dominance of *Firmicutes* in this age group, which could be due to organ specificity. This increased abundance of SCFA-producing bacteria in adolescent laryngeal microbiota may be indicative of differential metabolic needs in the larynx. We found that metabolic functions were one of the represented functions differentially predicted by PICRUSt2 analysis. Specifically, we found that butyrate and propionate metabolism to be highest in the young adult mice (Fig 6B). SFCA-producing bacteria are responsible for maintaining epithelial barrier integrity, lipid and energy metabolism, immunological and inflammatory responses, as well as hemostasis [46]. Future studies should consider colonizing germ-free mice with these SFCA-producing bacteria to delineate their specific metabolic role in the larynx across throughout the lifespan.

**Table 2. Differentially represented functions predicted by PICRUSt2.**

| Feature | KEGG pathway (level-1) | KEGG pathway (level-2) | Effect size (Eta-squared) | Corrected p-value (Bonferroni) |
|---|---|---|---|---|
| Lysine degradation | Metabolism | Amino acid metabolism | 0.727 | 3.78E-03 |
| Tryptophan metabolism | | | 0.76 | 1.23E-03 |
| Valine, leucine and isoleucine degradation | | | 0.77 | 8.43E-04 |
| Phenylalanine metabolism | | | 0.678 | 1.60E-02 |
| Amino sugar and nucleotide sugar metabolism | | Carbohydrate metabolism | 0.648 | 0.036 |
| Citrate cycle [TCA cycle] | | | 0.678 | 1.60E-02 |
| Fructose and mannose metabolism | | | 0.645 | 3.90E-02 |
| Galactose metabolism | | | 0.665 | 2.30E-02 |
| Inositol phosphate metabolism | | | 0.752 | 1.60E-03 |
| Pentose phosphate pathway | | | 0.712 | 6.03E-03 |
| Carbon fixation in photosynthetic organisms | | Energy metabolism | 0.669 | 2.10E-02 |
| Carbon fixation pathways in prokaryotes | | | 0.661 | 2.60E-02 |
| Sulfur metabolism | | | 0.75 | 1.75E-03 |
| Sulfur relay system | | | 0.71 | 6.41E-03 |
| Biosynthesis of unsaturated fatty acids | | Lipid metabolism | 0.732 | 3.19E-03 |
| Fatty acid metabolism | | | 0.788 | 4.03E-04 |
| Synthesis and degradation of ketone bodies | | | 0.712 | 6.04E-03 |
| Biosynthesis of siderophore group nonribosomal peptides | | Metabolism of terpenoids and polyketides | 0.658 | 0.028 |
| Carotenoid biosynthesis | | | 0.698 | 9.14E-03 |
| Geraniol degradation | | | 0.703 | 7.98E-03 |
| Zeatin biosynthesis | | | 0.737 | 2.70E-03 |
| Biotin metabolism | | Metabolism of cofactors and vitamins | 0.661 | 0.025 |
| Benzoate degradation | | Xenobiotics biodegradation and metabolism | 0.682 | 0.014 |
| Butyrate metabolism | | | 0.752 | 1.61E-03 |
| Caprolactam degradation | | | 0.765 | 9.93E-04 |
| Glyoxylate and dicarboxylate metabolism | | | 0.64 | 4.40E-02 |
| Propionate metabolism | | | 0.719 | 4.93E-03 |
| Styrene degradation | | | 0.739 | 2.51E-03 |
| Tropane, piperidine and pyridine alkaloid biosynthesis | | Biosynthesis of other secondary metabolites | 0.769 | 8.65E-04 |
| Cell cycle—Caulobacter | Cellular Processes | Cell growth and death | 0.73 | 3.49E-03 |
| Peroxisome | | Transport and catabolism | 0.699 | 8.90E-03 |
| RNA transport | Genetic Information Processing | Translation | 0.757 | 1.38E-03 |

*Lactobacillus*, *Prevotella*, *Bifidobacterium*, *Ruminococcus*, and *Akkermansia* were indicator species for adolescent mice, which is not surprising given that these bacteria are abundant in younger individuals [15, 16]. Interestingly, we found that *Actinomyces* and *Veillonella* were indicator species for young adults. Abundance of these bacteria in the gut have been associated with unhealthy aging (frailty, inflammaging etc.) [11, 12]. The presence of these bacteria–as indicator species–for adolescent and young adult mice may represent the distinct physiological and cellular functions in the larynx (i.e., organ specificity). Cellular process (cell growth and death) is a represented function for KEGG pathways represented by PICRUSt2. More research is required to identify functional consequences of these KEGG pathways functions across life-span (including translation).

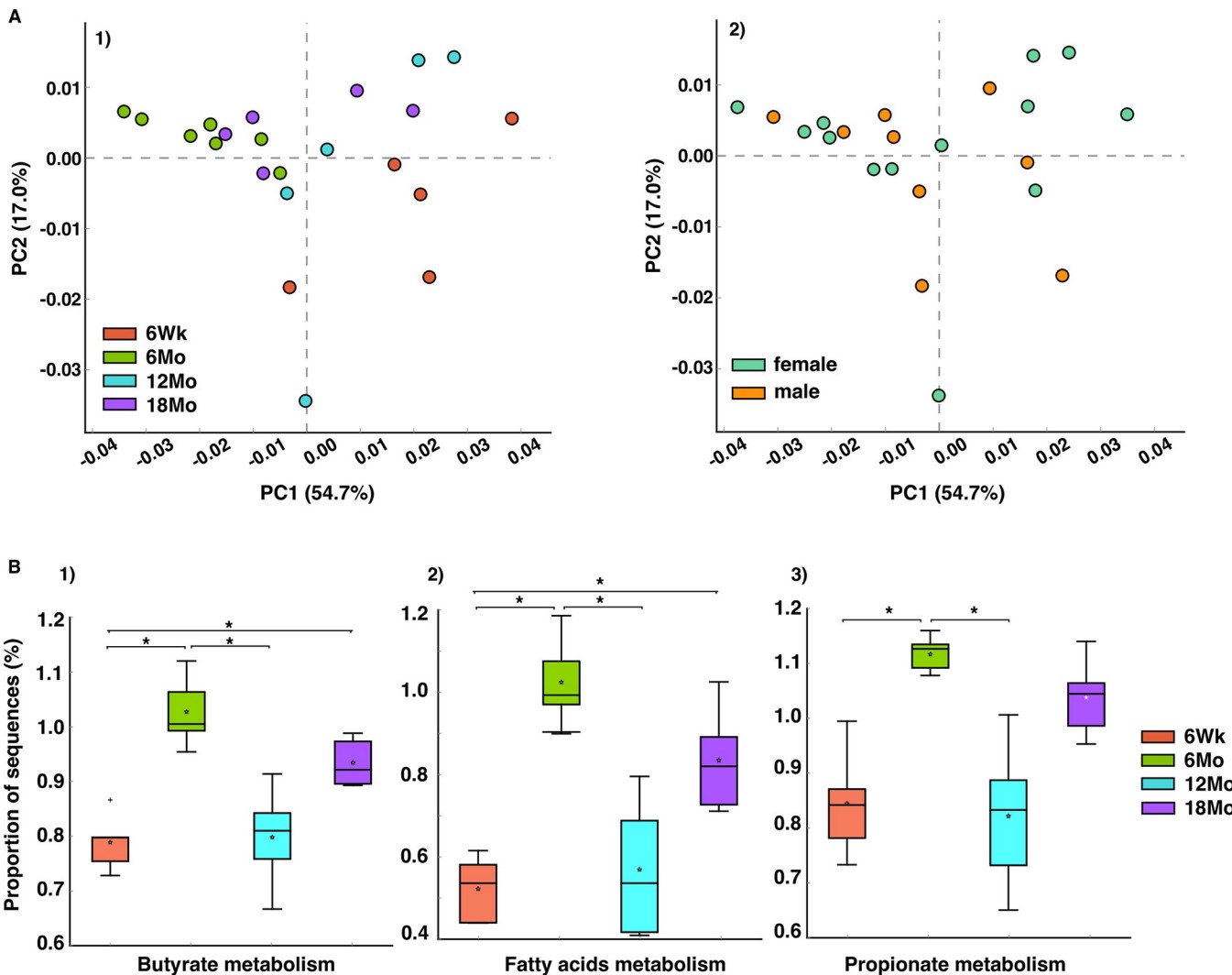

**Fig 6. Differentially represented metagenomics functions across age groups predicted by PICRUSt2.** (A) Principal Coordinate Analysis (PCoA) of KEGG ortholog (KOs) abundances determined via PICRUSt2 analysis showing the dissimilarity of predicted metagenomics of the laryngeal bacterial communities in each mouse group; each dot represents a laryngeal sample. (B) Boxplot showing the proportion of differentially represented KOs abundances across age groups. Statistical analyses were done by one-way ANOVA and Tukey HSD post-hoc comparisons across age groups; "*" above the black line represents the differentially abundant taxa were statistically significant ($p < 0.05$). Upper whisker, lower whisker, black line in the box, "*" inside the box, "+" represents the maximum, minimum, median, mean, and outlier of the data. 6Wk = adolescent, 6Mo = young adult, 12Mo = older adult, 18Mo = elderly.

In young adults, we found that *Fusobacterium* is an indicator species. The prevalence of this bacteria increases with age in the gut [11, 12]. In young adults, we found the significant enrichment of *Corynebacterium*. *Corynebacterium* generally requires moist environments, such as sweat and mucus, for colonization [14, 47]. The significant enrichment of *Corynebacterium* in young adults is an interesting finding, as this bacteria can be associated with lower levels of allergen sensitization in the nasal microbiome [48].

We also found enrichment of *Actinobacteria* and *Proteobacteria* specific to young and older adults. Whereas, *Actinobacteria* are essential for modulation of epithelial permeability and immunoregulation [49]; *Proteobacteria* maintain an ideal environment for anaerobic colonization by regulating oxygen levels [50]. Healthy age-related changes are associated with decreased abundance of *Bacteroides* with age [39]. However, we found a significant variation

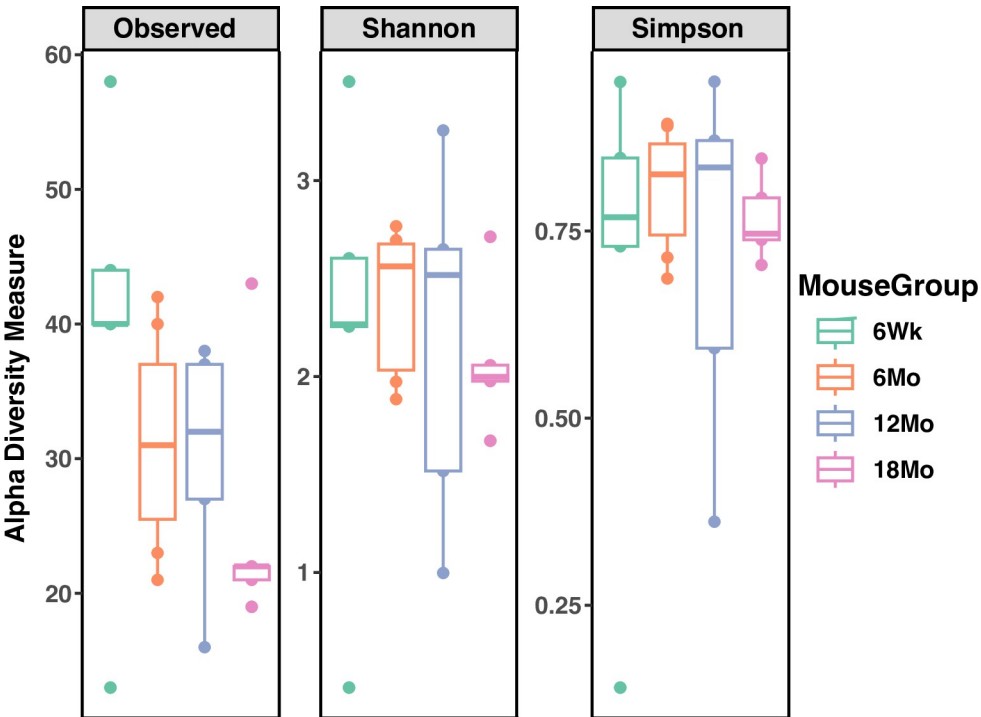

**Fig 7. Comparison of alpha diversity metrics within age groups.** 6Wk = adolescent, 6Mo = young adult, 12Mo = older adult, 18Mo = elderly.

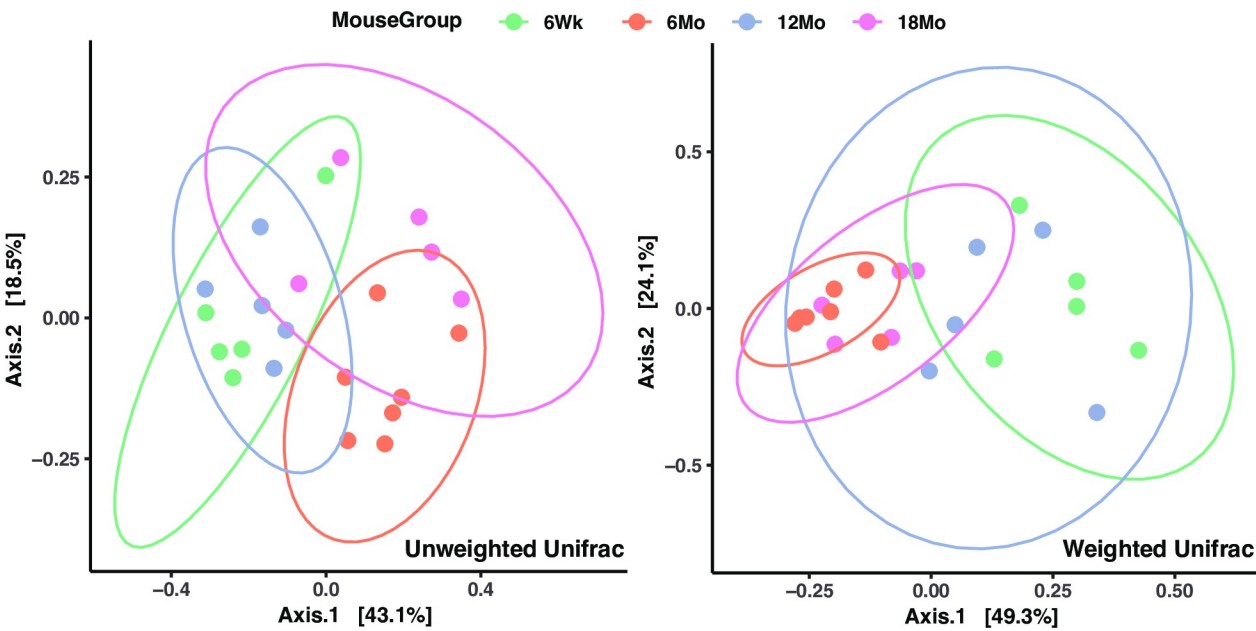

**Fig 8. Beta diversity analysis assessing microbial community structure across age groups.** Principle Coordinate Analysis (PCoA) of unweighted (left) and weighted (right) UniFrac distance. Each dot represents a mouse laryngeal sample; contours indicate the groups obtained by comparisons with PERMANOVA ($p$ = 0.162, F = 1.8664). 6Wk = adolescent, 6Mo = young adult, 12Mo = older adult, 18Mo = elderly.

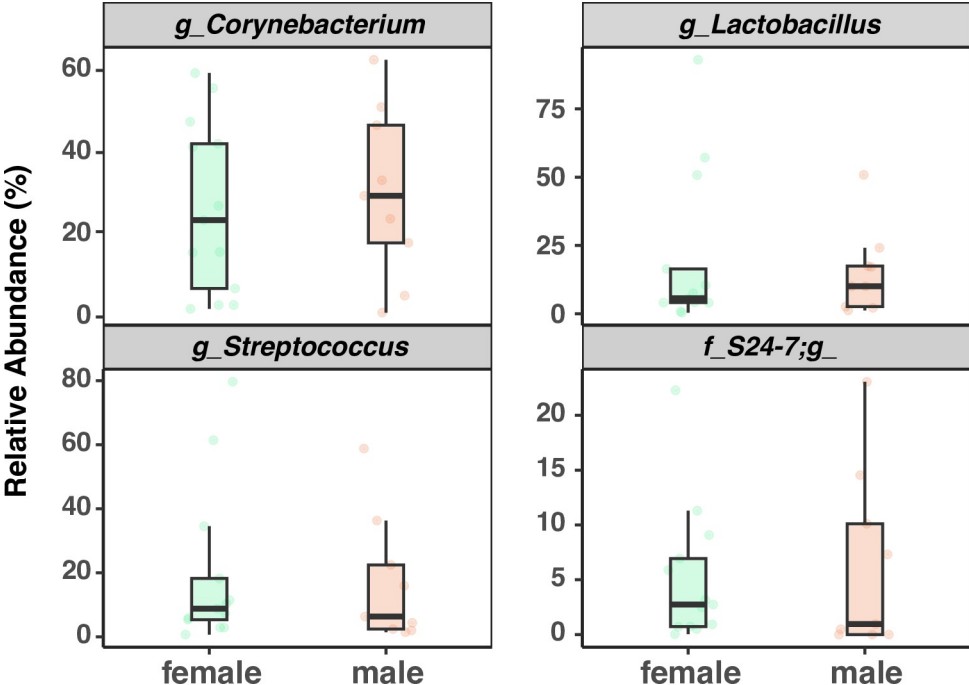

**Fig 9. Comparison of the relative abundance of top 4 genera across sexes.** g, f represents genera and family. 6Wk = adolescent, 6Mo = young adult, 12Mo = older adult, 18Mo = elderly. Statistical analyses were done by one-way ANOVA (*p* = 0.05) and Tukey HSD post-hoc comparisons across sex groups.

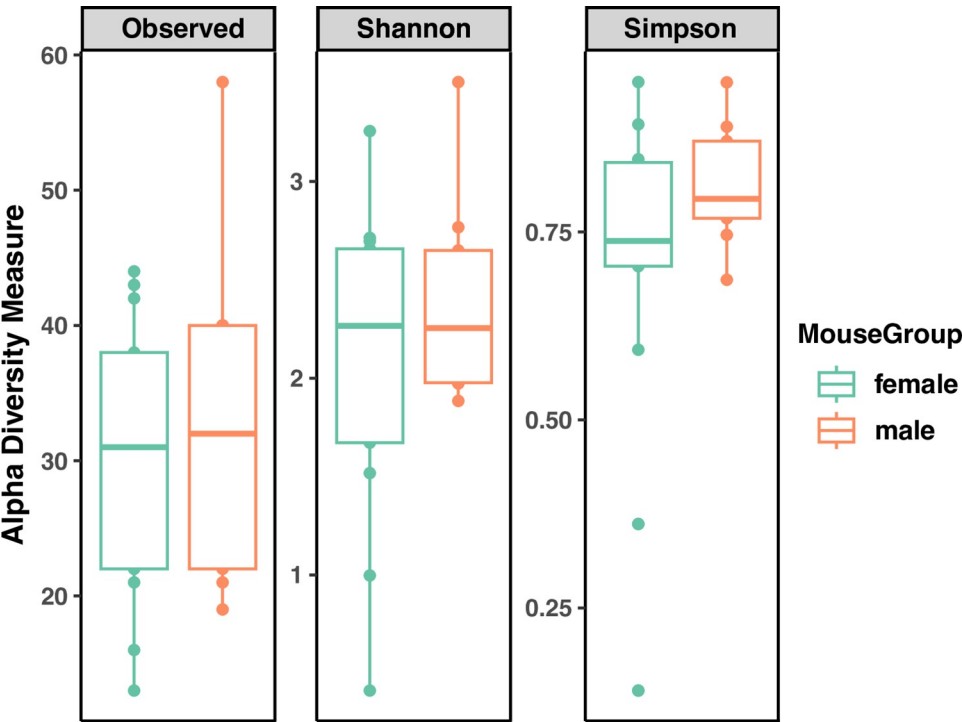

**Fig 10. Comparisons of alpha diversity metrics within biological sex.** 6Wk = adolescent, 6Mo = young adult, 12Mo = older adult, 18Mo = elderly.

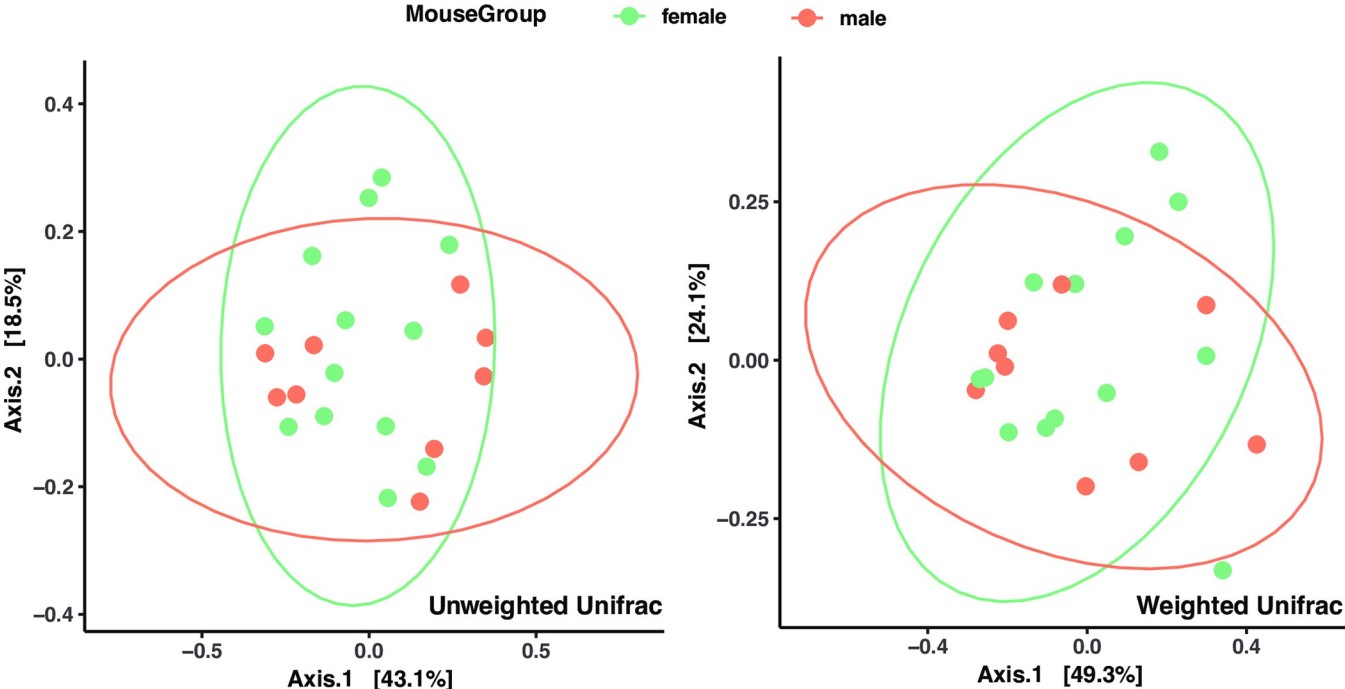

**Fig 11. Beta-diversity analysis showing difference in microbial community structure across biological sexes.** Principle Coordinate Analysis (PCoA) of unweighted (left) and weighted (right) UniFrac distance. Each dot represents a mouse laryngeal sample; contours indicate the groups obtained by comparisons with PERMANOVA ($p > .005$). 6Wk = adolescent, 6Mo = young adult, 12Mo = older adult, 18Mo = elderly.

of *Bacteroides* across age groups (i.e., decreased abundance of *Bacteroides* in young adults compared to adolescents, followed by an increase in older adults and elderly). The aging murine gut is also associated with a "trade-off" of other *Firmicutes* genera for *Clostridia*, in later life [13].

While we found age-specific taxonomic variation, there was relative stability of laryngeal microbiota within each age group (largest microbial variation observed in adolescents across all measures of alpha diversity). This may be indicative of high laryngeal resistance to colonization that stabilizes post-development [51]. Similarly, oropharyngeal microbiota is reported priorly to have similar temporal stability across age groups [19]. We found no significant separation in the laryngeal microbiota (i.e., no significant differences in post-hoc analyses of beta-diversity) across age groups. Although the gut microbiota has been shown to vary throughout the lifespan [52], similar result has not been reported in the laryngeal microbiome. Lastly, while the selection of age group is based on prior research [27], it is possible that the age groups may not accurately reflect microbial variation across aging.

A secondary aim of our study was to explore potential sex-related variation in laryngeal microbiota. Although sex hormones can significantly affect alpha and beta-diversity in the gut microbiome [20, 21], there were no changes in our measures of alpha or beta-diversity, no differentially abundant taxa, nor changes in relative abundance in the top 4 genera between sexes. Possible reasons may account for the lack of observed laryngeal microbiota variation. Sex hormones are time and age-dependent [21]. We assumed that sex-induced variation in laryngeal microbiota may be confounded by collapsing of age groups. However, differential abundance analysis using *limma-voom function* suggests the confounding age effects are negligible in our case. Reports of sex-related variation in the gut microbiome remain mixed [53–55], and independent microbial effects of sex may not be as apparent in an organ with relatively low

bacterial load [such as the larynx] [25]. Although female mice were housed together, we did not perform estrus staging. Thus, hormonal variation in females across different age groups could confound potential sex-related variation in laryngeal microbiota across the lifespan. Sex-hormones are thought to affect microbiota in the gut [56].

## Limitations and future directions

Considering that *Streptococcus* is underrepresented in positive controls of the oral microbiome whole cell mix, which is likely due to the potential bias of the current sample processing protocol towards certain groups of microorganisms, the relative abundance of *Streptococcus* in all age groups could be underestimated. Before sex-related variation in the laryngeal microbiota can be discounted, future studies with a larger sample size of each sex, and multiple age groups are needed for corroboration (n > 4). Although changes of SCFA-producing bacteria were observed in the present study, their functional consequences on host laryngeal physiology remain unknown. Future studies should consider the colonization of germ-free mice with these SCFA-producing bacteria to delineate their specific metabolic role in the larynx across the age-span. In addition, murine laryngeal microbiota exhibited relative stability with age and sex, as seen within the oropharyngeal microbiota [19]. However, environmental differences, and mechanical stresses from vibration, differentiate the human model from the murine model. These factors may contribute to aging related findings and cannot be replicated in the murine model. Thus, future work should investigate whether similar age and sex-related variation exist in the human laryngeal microbiota to better understand the role of the microbiota in developing laryngeal pathologies.

## Conclusion

The laryngeal microbiota experiences variation as a function of age. There is a compositional shift in the laryngeal microbiota that occurs between adulthood and older adults. Differentially abundant phyla, such as *Firmicutes*, *Bacteroides*, *Lactobacillus*, and *Corynebacterium*, across the age-span may be indicative of the differing metabolic and immunological needs and differing colonization resistance of laryngeal microbiota compared to that of other body organs, such as the gut, nose, and oral cavity. Future studies should elucidate the role of specific bacteria in age-related decline in laryngeal function. Additionally, in our cohort of murine samples, there were no sex-related changes in the laryngeal microbiota.

## Acknowledgments

We sincerely thank the University of Wisconsin-Madison Biotechnology Center DNA Sequencing Facility for providing next generation sequencing services.

## Author Contributions

**Conceptualization:** Ran An.

**Data curation:** Ran An, Anumitha Venkatraman.

**Formal analysis:** Ran An, Anumitha Venkatraman.

**Funding acquisition:** Federico E. Rey, Susan L. Thibeault.

**Investigation:** Ran An, Anumitha Venkatraman, John Binns, Callie Saric.

**Methodology:** Ran An, John Binns, Callie Saric.

**Project administration:** Ran An, Federico E. Rey, Susan L. Thibeault.

**Resources:** Federico E. Rey, Susan L. Thibeault.

**Software:** Federico E. Rey.

**Supervision:** Federico E. Rey, Susan L. Thibeault.

**Visualization:** Ran An, Anumitha Venkatraman.

**Writing – original draft:** Anumitha Venkatraman.

**Writing – review & editing:** Ran An, Federico E. Rey, Susan L. Thibeault.

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
