## [Decision Letter · Decision Letter 0]

16 Jan 2024

PONE-D-23-38789Age and Sex-related Variations in Murine Laryngeal MicrobiotaPLOS ONE

Dear Dr. Thibeault,

Thank you for submitting your manuscript to PLOS ONE. After careful consideration, we feel that it has merit but does not fully meet PLOS ONE’s publication criteria as it currently stands. Therefore, we invite you to submit a revised version of the manuscript that addresses the points raised during the review process.

We look forward to receiving your revised manuscript.

Kind regards,

Peter Gyarmati

Academic Editor

PLOS ONE

Journal Requirements:

3. To comply with PLOS ONE submissions requirements, in your Methods section, please provide additional information regarding the experiments involving animals and ensure you have included details on (1) methods of sacrifice, (2) methods of anesthesia and/or analgesia, and (3) efforts to alleviate suffering.

"SLT

R01DC012773-10 

NIDCD NIH

https://www.nidcd.nih.gov/

No"

6. Please expand the acronym “NIDCD NIH” (as indicated in your financial disclosure) so that it states the name of your funders in full.

7. When completing the data availability statement of the submission form, you indicated that you will make your data available on acceptance. We strongly recommend all authors decide on a data sharing plan before acceptance, as the process can be lengthy and hold up publication timelines. Please note that, though access restrictions are acceptable now, your entire data will need to be made freely accessible if your manuscript is accepted for publication. This policy applies to all data except where public deposition would breach compliance with the protocol approved by your research ethics board. If you are unable to adhere to our open data policy, please kindly revise your statement to explain your reasoning and we will seek the editor's input on an exemption. Please be assured that, once you have provided your new statement, the assessment of your exemption will not hold up the peer review process.

8. Please include your full ethics statement in the ‘Methods’ section of your manuscript file. In your statement, please include the full name of the IRB or ethics committee who approved or waived your study, as well as whether or not you obtained informed written or verbal consent. If consent was waived for your study, please include this information in your statement as well.

Reviewers' comments:

Reviewer's Responses to Questions

**Comments to the Author**

1. Is the manuscript technically sound, and do the data support the conclusions?

Reviewer #1: Yes

Reviewer #2: Yes

2. Has the statistical analysis been performed appropriately and rigorously? 

Reviewer #1: Yes

Reviewer #2: Yes

3. Have the authors made all data underlying the findings in their manuscript fully available?

Reviewer #1: Yes

Reviewer #2: Yes

4. Is the manuscript presented in an intelligible fashion and written in standard English?

Reviewer #1: Yes

Reviewer #2: Yes

5. Review Comments to the Author

Reviewer #1: This manuscript by An et al. describe the results of a study aiming to identify age- and sex-based differences in murine laryngeal microbiota. The authors used 16s rRNA sequencing to compare laryngeal microbial profiles between sexes and across four age cohorts. They compared relative abundances at the phyla and genera levels as well as measured a series of diversity indices and found differences across age, but not sex.

I found this study interesting and well designed. The methods were described in appropriate detail. Most of my comments are about how the study is presented and with significant revisions, I think that this study is publishable. Below I describe my overall concerns and then list some detailed comments.

1. Expanding and clarifying the results and figures

The figure legends were quite short and missing some information that would help readers. Across all the figures, it would be helpful if the authors included the age category names that associate with each group (e.g., 6 wk = adolescent) since they mainly refer to these names in the main text. It would also be helpful if there were a consistent color palate across figures for the age categories, but this is pretty minor and optional. Adding significance labels to the relevant figures (e.g., figure 4) would also help readers quickly identify the pattern the authors are pointing out in the text. Figure 5 and 6 would definitely benefit from further explanation. For example, in figure 6, what does the thick red vs thick black vs thin black lines mean on the plot? The results in Figure 6B are also not mentioned in the main text of the results (although I see they are discussed in the discussion). These data should be reported in the results section. I also find the reporting of the statistics in the results to be a little sparse. For example, on L204 the authors should mention what type of statistical test they used. I also noticed that the figure 1 labels are incorrect and do not match the description in the main text.

2. detecting sex-based differences

As the authors mention in the discussion, their ability to detect sex-based differences in laryngeal microbiota is likely confounded by age. Was there an attempt to account for this in the analyses? I imagine it is possible to fit a linear model to the data and remove the component that is due to the age effect. I believe there are functions within the limma R package that do this.

3. Adding to the discussion

The authors’ discussion focuses on contextualizing their results with what is known about gut microbiota and aging, but what about other studied organs? The oral cavity for example is connected to the larynx via the pharynx. Do the authors expect that certain organs might have more similar microbiomes? Also, I suggest adding a paragraph on the drawback of this study and/or future directions arising from the results of the authors’ work here.

4. Revising manuscript language to improve readability

I found this manuscript a little hard to follow and noticed some consistent typos throughout. I’ll provide a few examples here and highlight a couple of line edits below to use as a guide for revision. In the introduction the authors’ discussion of what is known about the gut microbiome (L86-90) was confusing. I found the methods generally very easy to follow. The results and discussion had a few typos including double parenthesis, using periods instead of commas, and sentences with missing words (e.g., L296 for a sentence with a missing word).

Detailed comments:

L96-97 – I don’t know that this is likely to be universally true across species. Be more specific here (murine models + humans?)

Methods – include versions for any software used where relevant.

L167-168 – Authors report 2 samples removed from adolescent group, but in the figures in the results it looks like 1 adolescent and 1 mature adult (6 wk and 12 mo) were removed. Please update whichever is incorrect.

L184 – write out amplicon specific variant fully at the first mention in this new section.

L186-188 – double parentheses here.

Results – Age groups should be referenced in a consistent manner. In paragraph 1 of results starting on L181 the authors call the same group “middle-aged” and then “older adult.”

L189-190 – move to discussion.

L203-204 – Is the significant decreased referring specifically to Corynebacterium?

L205 – Does “groups” refer to sex here?

L207-208 – stats are vague here.

L211 – write out what LEFSe is.

L248 – this is interpretation and should be in the discussion.

L318 – I wouldn’t say confirmed here. Why would we assume the same in the larynx?

L367 – A citation for a manuscript in review should not include the journal name.

Data availability statement should be written out in the manuscript by the acknowledgements.

Reviewer #2: - The authors state that that age related microbial variation may potentially underlie biological mechanisms and leave the larynx more susceptible to disease pathogenesis. The authors go on to discuss age related pathology, presumably presbylarynx. It would be beneficial to better understand how microbial changes could potentially drive the atrophy seem in this condition.

- While gut and skin changes are briefly introduced, the introduction would benefit from description of whether there has been studied age related changes in microbiota composition in other parts of the airway.

- While I understand the need to parse out individual microbial mechanisms, is is possible that the stability of the laryngeal microbiota was related to the chosen model and not actually reflected in humans? For example, the environment, mechanical stresses of vibration are needed to potentially interact and produce age related findings. There are some model limitations that should be discussed.

- Several proofreading errors were present in the manuscript.

6. PLOS authors have the option to publish the peer review history of their article (what does this mean?). If published, this will include your full peer review and any attached files.

Reviewer #1: No

Reviewer #2: No

---

## [Author Response · Author response to Decision Letter 0]

13 Feb 2024

January 17th 2024

Peter Gyarmati

Dear Dr. Gyarmati, 

 We would like to thank the reviewers for the valuable feedback on our manuscript. We are pleased that the reviewers found this manuscript “well-designed” and “interesting.” In incorporating reviewer feedback, we believe that the impact, organization and clarity of this manuscript is much improved. We are thankful for the opportunity to revise and resubmit. Please find our responses to each of the editor and reviewer’s comments below. 

Clarification of Funding and Funder role 

As per the editor’s request, we would like to clarify that “the funders had no role in study design, data collection and analysis, decision to publish, or preparation of the manuscript.” Our study was funded by the National Institute of Deafness and other Communication Disorders, National Institute of Health (NIDCD, NIH). We have made sure that funding information in the ‘Funding Information’ and ‘Financial Disclosure’ sections match.

Editor 

1. Asks authors to reformat manuscript and figures to PLOS ONE requirements.

RESPONSE: We have complied.

2. Asks authors to deposit original data in a repository. 

RESPONSE: 16S rRNA gene sequencing data are accessible in GenBank under BioProject PRJNA1041457.

3. Requests authors to add details to methodology regarding (1) methods of sacrifice, (2) methods of anesthesia and analgesia and (3) efforts to alleviate suffering. 

RESPONSE: Animals did not undergo any experimental procedure, thus, anesthesia, analgesia or measures to alleviate suffering were not employed in the present study. We have included additional details for our euthanasia procedure as follows; For the experiment, mice were sacrificed via CO2 inhalation in L103. 

4. Requests clarification of data sharing plan.

RESPONSE: As stated above, 16S rRNA gene sequencing data has been deposited in GenBank under BioProject PRJNA1041457 and will become available on acceptance. We also included a Data availability statement in the manuscript, L397. 

5. Add an ethics statement to the Methods section of the manuscript.

RESPONSE: We have complied. We have clarified our ethical review board approval with the following statement “All procedures and protocols were approved by the Institutional Animal Care and Use Committee (IACUC) at University of Wisconsin-Madison (Protocol: M005669).

This can be found in L94. 

Reviewer 1

We thank the review for the insightful comments. We have made changed accordingly. Please note that line number may have changed for some comments after revision. 

1. Suggests expansion and clarification of results and figures. 

RESPONSE: We thank the reviewer for this suggestion. We have incorporated the following changes based on the reviewer’s suggestion: (1) age category names that are associated with each group, (2) Figure 4 was edited to incorporate significant labels (3) Figures 5 and 6 now have further explanations, (4) added Figure 6B results to the main text (L210-213), (5) expanded on the statistical tests incorporated, (6) edited Figure 1 to match the text. 

2. Asks for clarification for the analyses for sex-based differences (to remove confounding age effects.

 RESPONSE: We have incorporated a Lar model with the data (using the limma-voom function in limma package in R) to remove the age effect and have included the corresponding results in the methods (L161), results (L275), and discussion (L361) of the manuscript. 

3. Requests additions to the discussion.

RESPONSE: We have incorporated all the reviewer’s suggestions. Specifically, we have added a paragraph describing the study limitations and future directions (L368). We have also age-related changes in the upper airway in the discussion (L347). 

4. Suggests the following changes to improve readability. 

1) Correct typos (including double parenthesis, using period instead of commas) and clarify information about the gut microbiome in the introduction.

RESPONSE: We have corrected all typo throughout. We have altered our discussion of the gut microbiome in the introduction to increase clarity (L57-67).

2) L296 has a missing word.

RESPONSE: We have corrected this sentence.

3) L96-97 requires clarification on the species.

RESPONSE: We have complied.

4) Methods include versions of software when relevant.

RESPONSE: version has been added to each software used in the manuscript. 

5) Correct L167-168 to show that 1 adolescent and 1 mature adult was removed.

RESPONSE: We have corrected this sentence, and now it’s in L153.

6) Incorporate the full form of ASV (amplicon sequence variant) at the start of the new section.

RESPONSE: We have complied, which is in L144.

7) Remove double parenthesis in L186-188.

RESPONSE: We have complied.

8) Correct paragraph 1 of the results (L181) as authors call the same group two names (“middle-aged”, “older adult”).

RESPONSE: We have complied and corrected L185 (line number changed after revision) to address this group consistently as “older adults”

9) Move L189-190 to the discussion. 

RESPONSE: We have complied.

10) Clarify whether L203-204 refers specifically to Corynebacterium. 

RESPONSE: We have complied (L204).

11) Clarify whether “groups” in L205 refers to sex. 

RESPONSE: We have complied.

12) Clarify the statistics in L207-208. 

RESPONSE: We have complied. We add more detailed statistics to the Methods section (Welch’s t-test, L166). 

13) Write out the full form of LEFSe, 

RESPONSE: We have complied.

14) Move L248 to the discussion.

RESPONSE: We have complied.

15) Remove the word “confirmed” in L318.

RESPONSE: We have complied.

16) Remove the journal name when citing a manuscript in review.

RESPONSE: We have complied.

17) Data availability statement should be placed with the acknowledgements.

RESPONSE: We have complied (L397).

Reviewer 2

1. Asks for clarification on how microbial changes could potentially drive atrophy in presbylarynx. 

RESPONSE: We thank the reviewer’s insightful comment. We agree that understanding how microbial changes could potentially drive the atrophy in presbylarynx or other diseased conditions is important and beneficial. As a first step towards the long-term goal, this study intends to delineate age-related variation in the resident microbiota (normal flora) in the larynx of conventionally raised mice. It is, to the authors’ knowledge, the first study to investigate the temporal stability of laryngeal microbiota across the lifespan in any model (murine or human). We have on-going projects in the lab along this line of research, where germ-free mice are colonized with a single or mixed bacterial strains to delineate how specific microbial variation influence host laryngeal physiology. 

2. Requests expansion of age-related changes of microbiota in other parts of the airway in the Introduction. 

RESPONSE: We appreciate this reviewer’s comment. Microbiota in other parts of the airway (oral and nasal) have a similar lack of literature delineating age-related variation. In the oral microbiota, Neisseria is reduced, and Streptococcus anginosus and Gemella sanguinis are increased with age (Kazarina et al., 2023), which are not observed in the laryngeal microbiota. Thus, more research is required to compare similarities and differences in all upper airway microbiota across the lifespan. As found in the laryngeal microbiota, oral microbiota also show high variation in microbial composition across the lifespan (Bach et al., 2020). We have incorporated this in the manuscript, which can be found in L68, 71,74. 

3. Suggests an expansion of model limitations (specifically the lack of mechanical stresses from vibration and the environment that exist in the murine model)

RESPONSE: We have added the following to the limitations section as per this reviewer’s suggestion; “In addition, murine laryngeal microbiota exhibited stability with age and sex. However, we acknowledge that environmental differences, as well as mechanical stresses from vibration, differentiate the human model from the murine model. These factors may contribute to aging related findings and cannot be replicated in the murine model. Thus, future work should investigate whether similar age and sex-related variation exist in the human laryngeal microbiota. We have addressed these limitations and next steps in the manuscript”. This can be found in L378. 

4. Asks for correction of typos.

RESPONSE: We have carefully revised the manuscript and corrected all typos.

---

## [Decision Letter · Decision Letter 1]

4 Mar 2024

Age and Sex-related Variations in Murine Laryngeal Microbiota

PONE-D-23-38789R1

Dear Dr. Thibeault,

We’re pleased to inform you that your manuscript has been judged scientifically suitable for publication and will be formally accepted for publication once it meets all outstanding technical requirements.

Kind regards,

Peter Gyarmati

Academic Editor

PLOS ONE

Additional Editor Comments (optional):

Reviewers' comments:

Reviewer's Responses to Questions

**Comments to the Author**

1. If the authors have adequately addressed your comments raised in a previous round of review and you feel that this manuscript is now acceptable for publication, you may indicate that here to bypass the “Comments to the Author” section, enter your conflict of interest statement in the “Confidential to Editor” section, and submit your "Accept" recommendation.

Reviewer #1: All comments have been addressed

Reviewer #2: All comments have been addressed

2. Is the manuscript technically sound, and do the data support the conclusions?

Reviewer #1: Yes

Reviewer #2: Yes

3. Has the statistical analysis been performed appropriately and rigorously? 

Reviewer #1: Yes

Reviewer #2: Yes

4. Have the authors made all data underlying the findings in their manuscript fully available?

Reviewer #1: Yes

Reviewer #2: Yes

5. Is the manuscript presented in an intelligible fashion and written in standard English?

Reviewer #1: Yes

Reviewer #2: Yes

6. Review Comments to the Author

Reviewer #1: (No Response)

Reviewer #2: The present study explored potential age and sex-related changes in the laryngeal microbiota across the lifespan in a murine model. The reviewer comments have been addressed and are incorporated ito the revised paper.

7. PLOS authors have the option to publish the peer review history of their article (what does this mean?). If published, this will include your full peer review and any attached files.

Reviewer #1: No

Reviewer #2: No
